# Prognostic Model of COVID-19 Severity and Survival among Hospitalized Patients Using Machine Learning Techniques

**DOI:** 10.3390/diagnostics12112728

**Published:** 2022-11-08

**Authors:** Ivano Lodato, Aditya Varna Iyer, Isaac Zachary To, Zhong-Yuan Lai, Helen Shuk-Ying Chan, Winnie Suk-Wai Leung, Tommy Hing-Cheung Tang, Victor Kai-Lam Cheung, Tak-Chiu Wu, George Wing-Yiu Ng

**Affiliations:** 1Allos Limited, 1 Hok Cheung Street, Kowloon, Hong Kong, China; 2Department of Physics, University of Oxford, Oxford OX1 3PJ, UK; 3Department of Physics, Fudan University, Shanghai 200433, China; 4Division of Infectious Diseases, Department of Medicine, Queen Elizabeth Hospital, Hong Kong, China; 5Division of Integrative Systems and Design, Hong Kong University of Science and Technology, Hong Kong, China; 6Multi-Disciplinary Simulation and Skills Centre, Queen Elizabeth Hospital, Hong Kong, China; 7Intensive Care Unit, Department of Medicine, Queen Elizabeth Hospital, Hong Kong, China

**Keywords:** COVID-19, triage, machine learning

## Abstract

We conducted a statistical study and developed a machine learning model to triage COVID-19 patients affected during the height of the COVID-19 pandemic in Hong Kong based on their medical records and test results (features) collected during their hospitalization. The correlation between the values of these features is studied against discharge status and disease severity as a preliminary step to identify those features with a more pronounced effect on the patient outcome. Once identified, they constitute the inputs of four machine learning models, Decision Tree, Random Forest, Gradient and RUSBoosting, which predict both the Mortality and Severity associated with the disease. We test the accuracy of the models when the number of input features is varied, demonstrating their stability; i.e., the models are already highly predictive when run over a core set of (6) features. We show that Random Forest and Gradient Boosting classifiers are highly accurate in predicting patients’ Mortality (average accuracy ∼99%) as well as categorize patients (average accuracy ∼91%) into four distinct risk classes (Severity of COVID-19 infection). Our methodical and broad approach combines statistical insights with various machine learning models, which paves the way forward in the AI-assisted triage and prognosis of COVID-19 cases, which is potentially generalizable to other seasonal flus.

## 1. Introduction

The coronavirus disease (COVID-19) pandemic, caused by the severe acute respiratory syndrome coronavirus 2 (SARS-CoV-2), started in Hubei province of China in December 2019 and has since spread worldwide, claiming the lives of more than 5 million people, as of January 2022 [1]. The virus and its variants, all possessing high-transmissibility properties, cause a variety of symptoms, ranging from acute respiratory distress to (systemic) organ failure [2]. Medical institutions and professionals all over the world have dealt with never-before-seen emergencies: overcrowded hospitals, scarce medical resources and response systems pushed to their limits. Although diagnostic tests, with variable sensitivity and specificity, have been widely available since 2020, it is still problematic to predict when a new peak of infection will present itself in a population and what measures should be taken to contain the spread while furnishing appropriate medical care. For these reasons, researchers have tried to identify specific features or test results that may be reasonably used as a predictor of the Severity of respiratory distress for COVID-19 positive patients as well as their risk of death [3,4,5,6,7,8,9,10,11,12,13,14] (see also [15] and reference therein).

We study the effect of various pre-treatment features on the final status (outcome) of COVID-19 patients in Hong Kong with the goal of facilitating a faster and more informed response program for triaging infected individuals. The aim of this triaging model is to accurately predict the risk of death or severe COVID-19 infection of a patient at the moment of admission. This way, the most severe infections would take priority in terms of medical care. As a preliminary step, we conduct a statistical analysis based on the computation of correlation coefficients and significance tests to determine the extent of influence of each feature on the patient triaging outcomes. We then perform feature selection using the mutual information as a measure to identify the most predictive features. Finally, the outputs of the feature selection process are passed to the machine learning (ML) model. We will comment on the methods used in our analysis and present the results, including the accuracies for our ML model. For completeness, we will also test the predictive power of our ML model when the full feature list is used in the categorization.

## 2. Methods

This is a retrospective observational study of 429 patients admitted between 12 February 2020 and 25 August 2020 to Queen Elizabeth Hospital under suspicion of COVID-19 infection. These patients were subsequently confirmed to be COVID-19 positive via polymerase chain reaction (PCR) testing and retained in the hospital for varying periods of time. The dataset includes information about the patients, among which their pre-existing medical conditions (comorbidities), test results at admission and amount of oxygen administered. During their stay in the hospital, these patients also received medical treatment based on their health condition and consequential medical suggestion. The nine treatments used were: interferon beta-1b, ribavirin, lopinavir/ritonavir (Kaletra), remdesivir, tocilizumab, dexamethasone, hydrocortisone, prednisolone and convalescent plasma.

Patients are labeled by one of two separate outcome results: Mortality and Severity. The Severity of the disease, related to the severity of respiratory distress, is defined based on the total amount of oxygen given to the patients during hospital stay: 0—stable case (0 L of oxygen/min); 1—mild case (1–3 L O_2_/min); 2—serious case (3–6 L O_2_/min); 3—critical case (>6 L O_2_/min). The outcome Mortality is self-explanatory and simply distinguishes patients who have survived from patients who have died (i.e., 0—alive; 1—dead), within a few months window of time.

An unbiased estimate of the effect of treatments on either outcome cannot be worked out from our dataset (see comments in Section 4); we instead present below a precise statistical analysis of the relevant discrete and continuous features, establishing which one(s) is most associated with either Severity or Mortality. In order to eliminate any spurious correlation between features, which may over- or under-estimate the power of certain features in predicting triage-outcomes; we will first test the association between features themselves.

We then consider a machine learning feature selection algorithm based on the mutual information between features and outcomes. The resulting features of this classifier are employed as input variables in well-known machine learning algorithms to predict the Severity and Mortality-chances for each patient.

### 2.1. Statistical Analysis for Feature Selection

To enhance the predictive power of the machine learning algorithm, and given the low number of data points present, we intend to select the subset of the full list of features which can best predict a triaging outcome.

We start by calculating the Spearman’s rank correlation coefficient ρ [16], which is particularly suited to extract the measure of monotonic association between discrete and/or continuous features. Next, we proceed by considering a natural distinction between discrete and continuous data and the study of their association with triaging outcomes, which merit separate discussions in the subsequent subsections.

#### 2.1.1. Discrete Features

At the moment of hospital admission, patients were asked to give background information about existent comorbidities and other categorical/discrete features: chronic heart disease, hypertension, asthma, chronic kidney disease, diabetes mellitus, hematological malignancies, sex and age. In addition to the feature “age”, which we bin into 5 sub-categories, the rest of these features are in fact binary, with 0 (1) representing, respectively, the lack (or presence) of a comorbidity or male (female) sex.

In order to extract the measure of association between these discrete features and the triaging outcomes, we use the well-known Pearson’s $\chi^{2}$-test of statistical significance [17] based on the $\chi^{2}$ test distributions (for more details, we refer the reader to the Section A.1).

#### 2.1.2. Continuous Features

The admitted patients also underwent blood-tests, from which a variety of continuous features values were extracted. Specifically, the features examined were: total bilirubin, ALT, creatine kinase, neutrophil, hsTnI (high-sensitivity troponin I), urea, CRP, Hb, WBC, lymphocyte, LDH, creatinine and platelet count. For each feature *f* in this list, we consider the 2 Mortality labels and the 4 Severity labels, each of which generates a distribution, $M_{0}^{f},M_{1}^{f}$ and $S_{0}^{f},S_{1}^{f},S_{2}^{f},S_{3}^{f}$, respectively. Here, we will focus on statistical point estimation, i.e., median and percentile ranges, so as to compare the statistical distributions of features of patients infected with COVID-19 with known statistical point estimation for normal-values. Specifically, we will make use of two different metrics which quantify the (normalized) “distance” between the median and interquartile range of the distribution of values of a feature, with the known non-infected population’s interquartile range. To be more precise, the first metric will measure how far the median of a certain distribution of COVID-19 infected patients falls from the normal population interquartile range. Similarly, the second metric will measure how much overlap there is between the interquartile range of a distribution of COVID-19 positive patients and the normal population interquartile range. We give the explicit mathematical form of these metrics in the Appendix A to this paper.

### 2.2. Machine Learning for Feature Selection

We employ a powerful feature selector, based on mutual information [18], which quantifies the extent of information gain due to a chosen feature, on the target outcome (see the Appendix A for details). The implementation of our classifier from scikit-learn, sklearn.feature_selection.mutual_info_classif, based on work by Ross [19] is specifically designed to deal with hybrid discrete and continuous features, as is the case in our analysis. The algorithm runs over the totality of the data, starting from an initial random seed. Given the small size of the dataset, the results vary slightly. We take an average of the results over 100 runs to decrease the bias of the selector.

### 2.3. Machine Learning for Triage-Prediction

We performed a comparative study of several conventional classifiers: Decision Tree, Random Forests [20], Gradient Boosting [21,22] and RUSBoost models [23]. The Decision Tree classifier utilises a max depth of 400 beyond which the tree does not branch out, while the Random Forest uses 100 estimators. The boosting models use 280 estimators for both RUSBoost and for Gradient Boosting. We choose this to be our parameter space after running multiple trials to maximize the accuracy while placing emphasis on reducing the number of false-negative predictions of the machine learning model in the Mortality class.

The model(s) utilizes a random 70–30% split between training and test data. Given the small sample and consequent large variances, the accuracy of algorithms will also vary within a few percentage points. This is what would one expect from an unbiased classifier trained on a small dataset (Algorithms trained on small samples of data producing constant results for complex classifications analyses are inevitably biased).

### 2.4. Data Cleaning

We imposed a large 25% tolerance on missing data, i.e., features with more than 25% unpopulated information were eliminated. For features with unpopulated data below this threshold, we draw synthetic representative data from the appropriate intervals corresponding to the outcome class. The feature for which the most entries were generated is hsTnI (30 synthetic data points).

To further cull the irrelevant features, we also considered a Variance Threshold test, with a tolerance of 1%: we simply eliminated all features whose values do not vary significantly within the tolerance (in this case, only 1% of the patients’ data for a specific features are allowed to vary from a fixed value). This also would necessarily imply that such features cannot be good predictors of either triaging outcome.

### 2.5. Data Augmentation

Since the raw data contain a significantly larger number of patients who did not display serious symptoms (or did not die) as opposed to those severely affected by COVID-19, we use SMOTE to address the problem of imbalanced classification [24]. The Synthetic Minority Oversampling TEchnique (SMOTE) is a data augmentation technique that generates artificially more members of the minority class(es). This method adds new information to our model as it interpolates between examples that are selected in a particular region of the feature distribution. We set the nearest neighbour parameter for SMOTE to 3, i.e., the algorithm looks for 3 nearest neighbors in the minority class, to form the convex hull from which the synthetic samples are drawn. It is important to note that this tunes our model to recognize very selective patient traits for both classification, i.e., inevitably the bias of the predictive algorithm (discussed in the sections below) is increased.

## 3. Results

Among the 429 patients, four patients were discharged against medical advice, so their data will not enter any of our analyses. For seven other patients, data about oxygen therapy were not present: in total, 418 patients are labeled based on the total amount of oxygen given to them during their hospital stay.

Similarly, among the 425 initial patients, 76 were transferred to other hospitals by necessity, and unfortunately, it has been not possible to contact them afterwards. Hence, 349 of the initial 425 patients are labeled by Mortality.

In Table 1 and Table 2, we show point estimation of the collected data, for each of the two label classes, Mortality or Severity.

Figure 1 shows the result of the Spearman’s correlation test applied between the features themselves, with no accounting for the outcome. As it immediate to see, none of the features seem to possess a high degree of monotonic association; the only exception is perhaps represented by neutrophil count and WBC, whose exact relation is however unknown to us. More complex measures of dependence between features could be probed if more data were available; hence, for the purpose of this paper, we will consider the features under examination to be statistically independent of each other, knowing that in the worst case the use of both neutrophil count and WBC as inputs of the machine learning algorithms may create light redundancies.

### 3.1. Discrete Feature Selection

In the following, we present a statistical analysis of the degree of association of the discrete features, namely diabetes mellitus, hypertension, chronic heart disease, chronic kidney disease, asthma, hematological malignancy, age and sex with the Severity and Mortality outcomes. The latter feature does not seem to be relevant to discriminate between either of the outcomes, as it is immediately evident from Table 1 and Table 2. Below, in Table 3, Table 4 and Table 5, we present the contingency tables for the categorization by Severity of the discrete features which do not satisfy the truth of the null hypothesis of no association between the features and outcomes, i.e., of the discrete features which are statistically associated to the Severity outcome.

The features diabetes mellitus, hypertension and age are found to be associated to the severity outcome, while the null-hypothesis holds true for chronic heart disease, chronic kidney disease, asthma, hematological malignancy. The results are summarized in Table 6.

Next, we consider the association between the discrete features and the Mortality outcome. This time, we find that the null hypothesis can be rejected for both chronic heart disease and age but not for the other discrete features. The results are shown in Table 7 and Table 8 and summarized in Table 9.

Hence, the feature age seems to have a high degree of association to both the outcome examined, while pre-existing conditions seem to be relevant only for one of the two.

### 3.2. Continuous Feature Selection

In this section, we present in a different and more precise form the results already shown in Table 1 and Table 2. In Figure 2, we show for each outcome, Mortality and Severity, and each class distribution $M_{i}$ and $S_{i}$, the median values (red/blue dots) and the interquartile range (red/blue vertical segments). The green horizontal lines instead correspond to the interquartile range of values associated to normal population ranges [25]. If either distance $r_{a}$, $r_{b}$ (see Appendix A) for any feature *f* class distribution is above the threshold of 15%, the feature *f* will be selected as being statistically relevant, since its median or interquartile range values are not included in or do not overlap significantly with the normal range (obtained from statistics on non-infected patients population), testifying how the feature itself seems to be associated to a COVID-19-infection.

For the Mortality status, the features selected are:CK ($M_{1}$: $r_{b}=118.7\%$);Creatinine ($M_{0}$: $r_{b}=15.6\%$; $M_{1}$: $r_{a}=15.6\%$, $r_{b}=53.3\%$);CRP ($M_{0}$: $r_{b}=325\%$; $M_{1}$: $r_{a}=1200\%$, *);WBC ($M_{1}$: $r_{b}=27.3\%$);Hb ($M_{0}$: $r_{b}=21.6\%$; $M_{1}$: $r_{a}=35.1\%$, $r_{b}=75.7\%$);hsTnI ($M_{1}$: $r_{a}=132.7\%$, $r_{b}=682\%$);LDH ($M_{0}$: $r_{b}=29.5\%$; $M_{1}$: $r_{a}=93\%$, *);Neutrophil ($M_{1}$: $r_{b}=63.4\%$);Urea ($M_{1}$: $r_{b}=50\%$).

Similarly, for the distribution of features in the classes of Severity, we select:CRP ($S_{0}$: $r_{b}=150\%$; $S_{1}$: $r_{a}=400\%$,*; $S_{2}$: $r_{a}=1175\%$, *; $S_{3}$: $r_{a}=1162.5\%$, *);Creatinine ($S_{0}$: $r_{b}=15.5\%$);Hb ($S_{0}$: $r_{a}=18.9\%$; $S_{1}$: $r_{b}=43.2\%$; $S_{2}$: $r_{b}=41.9\%$; $S_{3}$: $r_{b}=44.6\%$);hsTnI ($S_{2}$: $r_{b}=93.4\%$; $S_{3}$: $r_{b}=113.3\%$);LDH ($S_{1}$: $r_{a}=23.5\%$, $r_{b}=118.5\%$; $S_{2}$: $r_{a}=111\%$, *; $S_{3}$: $r_{a}=78\%$, *);Neutrophil ($S_{3}$: $r_{b}=57.9\%$).

In the above lists, we have made use of the asterisk symbol * to indicate that, for those distributions, the interquartile range was fully outside the normal ranges: in those cases, we presented only the $r_{a}$ ratio, which was sufficient to select the associated features as statistically significant.

Note that these results serve as an indicator of discriminatory features: another complementary test will be discussed in the section below.

### 3.3. Mutual Information Feature Selection

Figure 3 shows the results of the mutual information feature selector for both Mortality and Severity outcomes. The eight most important features selected by the algorithm are given in Table 10. Notice, aside from age (here treated as a discrete variable), only continuous variables are selected by the mutual information algorithm.

### 3.4. Comparison with Feature Selection Results from Literature

Up until now, the emphasis of our analysis lay on the prediction of the features, both discrete and continuous, most relevant for triaging patients by Severity and Mortality. As a way to confirm the validity of our results for feature selection, we note that age [8,11,12,13], CHD [4,11,12], CRP [12,13], neutrophil [4,13] and LDH [7] were also proven to be statistically significant features to predict the Mortality outcome, while age [14], CRP and LDH [3,5,9,14] (among many others) were found to be statistically significant for the Severity outcome. Furthermore, in [15], a variety of results from other studies are summarized, which possess a notable overlap with the results presented above for the statistically relevant features to predict both outcomes.

### 3.5. Machine Learning—Severity and Mortality Prediction

Keeping the trade-off between accuracy and bias, which depends not only on the type but also the number of features inputted, in this section, we present the results obtained from our machine learning classifiers used as a predictor for the patient outcomes. We present the maximum accuracy results of a single run of the models in Figure 4 and Figure 5, where we use raw data first and raw data balanced by the addition of synthetic data later. In these plots, the 10 most significant features have been chosen as an input of the classifiers.

In Table 11 and Table 12, we collect the accuracies of all four algorithms, Gradient Boosting (GB), Decision Tree (DT), Random Forest (RF) and RUSBossting (RUSB) for prediction of the outcomes Mortality and Severity, respectively, averaged over 100 runs with different 70–30% training–test data split. The same results are presented in Table 13 and Table 14, where Smote was used to balance the dataset. Those accuracies have been computed using as an input 6 and 10 of the most predictive features as well as the full set (17) of features. For the outcome Mortality, those are $f_{6}^{M}=$ [C.H.D., Age, Urea, hsTnI, CRP, Creatinine], $f_{10}^{M}=\;f_{6}^{M}\;\cup \;$ [LDH, Neutrophil, WBC, Hb]; for the outcome Severity, we have instead $f_{6}^{S}=$ [Hypertension, Diabetes Mellitus, Age, CRP, LDH, hsTnI] and $f_{10}^{S}=f_{6}^{S}\cup$ [Lymphocyte, Creatinine, Urea, Neutrophil].

The left-hand figures in both Figure 4 and Figure 5 show confusion matrices in terms of the raw numbers, while the right-hand figures are the normalized confusion matrices.

## 4. Discussion and Conclusions

We analyzed COVID-19 patient test results, taken shortly after hospitalization and their medical history, such as comorbidities record. A preliminary analysis showed that the preprocessed dataset was complete in its entries but displayed signs of heavy class imbalance that we resolved by synthetically augmenting the minority classes in the dataset using SMOTE. We discovered that CK, creatinine, CRP, WBCs, Hb, hsTnI, LDH, neutrophil and urea levels were among the important features correlated with the Mortality outcome while CRP, creatinine, Hb, hsTnI, LDH and neutrophil levels were associated with the degree of Severity (respiratory distress) of each patient. These results were reaffirmed by the mutual information feature selector, which also revealed patient age to be highly correlated with both triaging outcomes, and confirmed by comparison with the literature [3,4,5,6,7,8,9,10,11,12,13,14,15].

The fact that both outcomes are predicted to be associated with almost the same set of features (see Table 10, and the substantial overlap with the features extracted from the earlier statistical analysis on continuous features), attests to the correctness and consistency of our analysis and results. However, since the lists are not exactly the same, one must decide which set of features to consider as input for the machine-learning algorithm. A statistical analysis of the accuracies confirms that the use of a larger (union) rather than a smaller (intersection) set of features marginally increases the accuracy (5–10%+) of our triaging algorithm but most likely also the bias of the results. Specifically, the machine learning algorithms we use perform very similarly on both classification tasks and their predictions are stable against randomized choice of training sample. Furthermore, they perform extremely well, as their accuracies testify: without the use of SMOTE, i.e., considering the original dataset with unbalanced outcome labels, the maximal accuracy is in the range [95.5–96.2%] for Mortality, and [73.7–79%] for Severity (note that this range corresponds to about 25 (mis)classified cases especially in the $S_{2}$ label). The range of accuracies when synthetic data are used to balance the labels are instead [98–99.1%] for Mortality and [74.2–94.6%] for Severity. This considerably improves on previously attempted similar studies which had even access to larger datasets, e.g., [7,10,11,12,13,14].

The analysis above is in principle amenable to a causal treatment, whereby the variables of a problem are represented as nodes and their causal relation (if any) as a directed arrow [26]. In the graph-based causal language, a directed arrow typically stems from the initial test results and points both at the treatment employed and the final outcome of the patient. In our study, however, the treatment was given independently of the initial test result, which might allow us to neglect the arrow connecting the initial test result to treatment. Furthermore, we note that (un)conscious biases may emerge during the administration of a treatment based on the visible (but not recorded) symptoms of the patient. The same unrecorded symptoms may also have a direct effect on the recorded symptoms, possibly creating a confounding arc that provides an unobserved backdoor pathway from the initial test results and the treatment, irrevocably stopping us from computing an unbiased estimate of the direct causal effect of test-results on outcomes (mediated by treatments). While in principle, one can identify and even quantify these effects to make useful predictions, the small size of our dataset limited the scope of our analysis.

It is worth noting that our study does not consider cases of long COVID-19 which requires data collected over longer periods. Similarly, our dataset provides information about the severity of respiratory distress due to COVID-19 during the course of hospitalization but not other flu-like symptoms such as fever, cough, etc., or other more dangerous effects, such as organ failures.

Notwithstanding the limitations imposed by the data on our analysis, the techniques adopted for the statistical and machine learning analyses of the data have shed valuable insight on the most likely features to be associated with severe and lethal COVID-19 infections in the sample of patients considered.

We conclude this discussion by noting that our methodical analysis exploiting both statistical and machine learning techniques, here used for the prognosis of COVID-19 patients, is also potentially generalizable to other seasonal infections and future pandemic diseases.

## Figures and Tables

**Figure 1 diagnostics-12-02728-f001:**
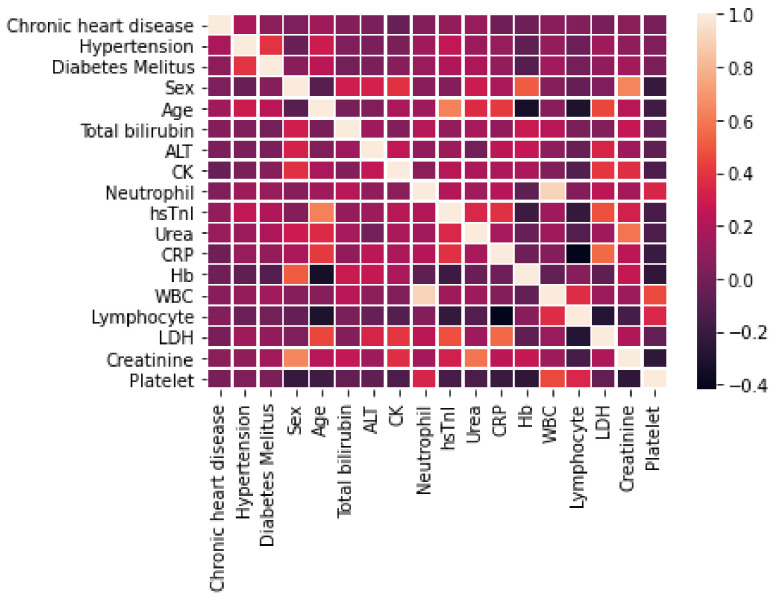
Spearman’s ρ—Feature to Feature.

**Figure 2 diagnostics-12-02728-f002:**
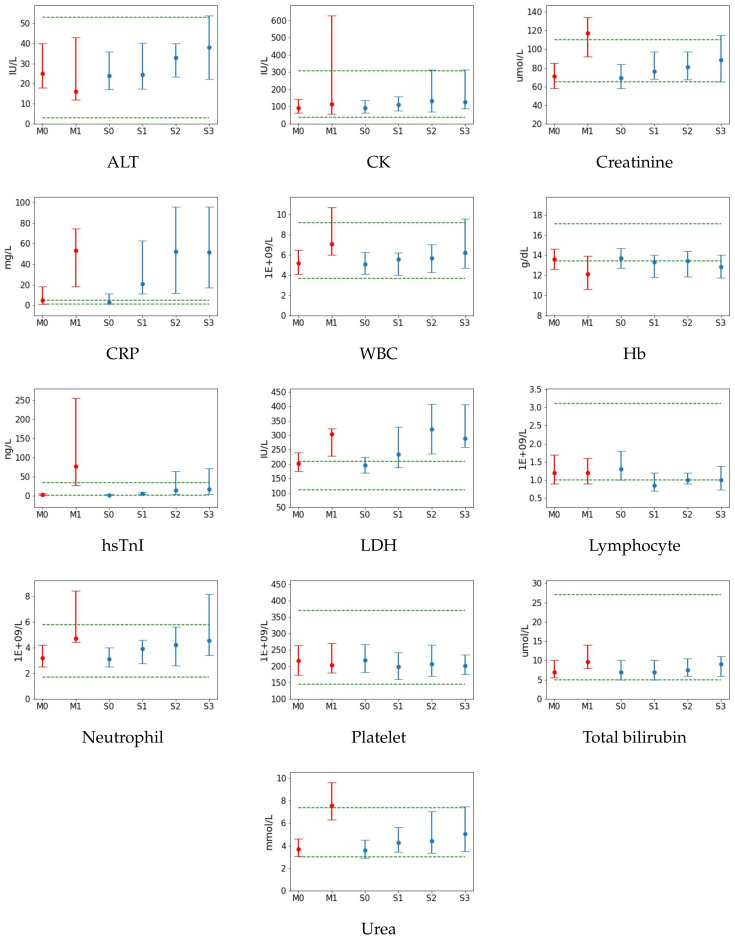
The median (dot) with the interquartile range (solid vertical lines) presented for each test result collected for patients categorized according to their Mortality (red) and Severity (blue) status. Dashed green lines represent the range of normality for the test result.

**Figure 3 diagnostics-12-02728-f003:**
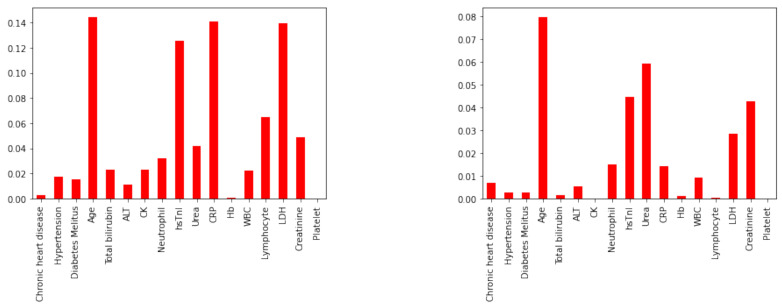
Mutual information classifier—Severity Outcome (**left**); Mortality Outcome (**right**).

**Figure 4 diagnostics-12-02728-f004:**
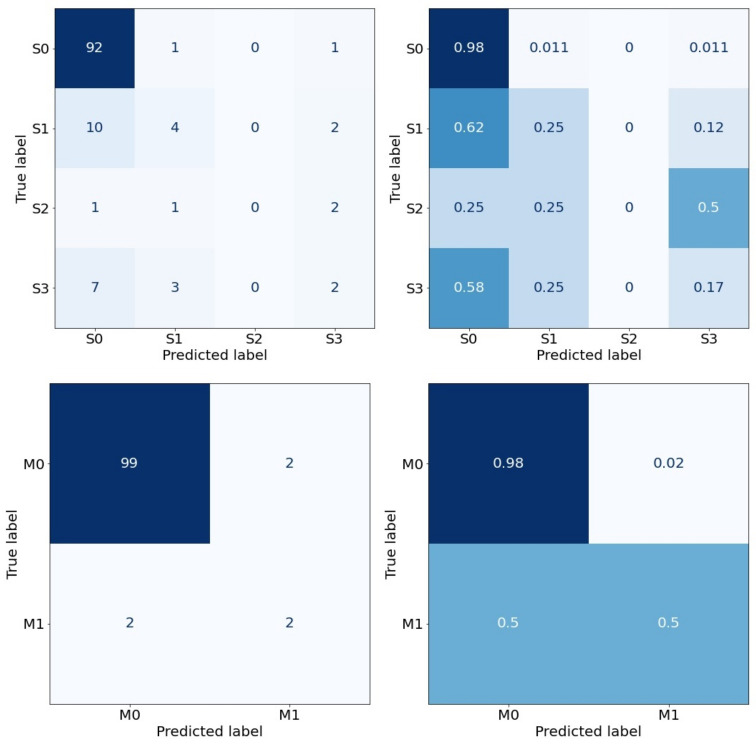
Random Forest classifier, Severity (**top**)—Accuracy: 77.8%. Gradient Boosting classifier, Mortality (**bottom**)—Accuracy: 96.2%.

**Figure 5 diagnostics-12-02728-f005:**
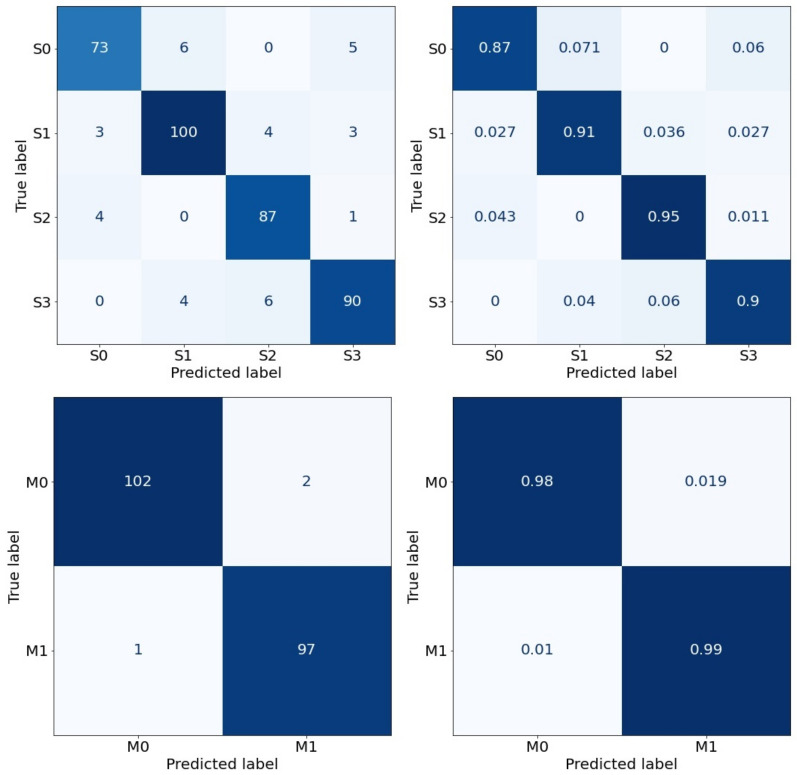
SMOTE—Random Forest classifier, Severity (**top**)—Accuracy: 90.7%. RUS Boosting classifier, Mortality (**bottom**)—Accuracy: 98.5%.

**Table 1 diagnostics-12-02728-t001:** The distribution of various features according to the binary patient outcome class Mortality. The number of patient comorbidities in each class is indicated as such (C.H.D: chronic heart disease; C.K.D.: chronic kidney disease; H.M.: hematological malignancies), while the median values and the interquartile range in parentheses are presented for each class (ALT: alanine aminotransferase, CK: creatine kinase, hsTnI: high sensitivity troponin I, CRP: C-reactive protein, Hb: hemoglobin. WBC: total white blood cell count, LDH: lactate dehydrogenase).

	Alive	Dead
Age	48.0 (16.71)	82.0 (11.31)
Sex		
Men	165	6
Women	171	7
Underlying disease		
C.H.D.	6	2
Hypertension	33	3
Asthma	1	0
C.K.D.	0	0
Diabetes Mellitus	25	0
H. M.	1	0
Test results		
Total bilirubin (umol/L)	7.0 (4.5)	9.7 (6.0)
ALT (IU/L)	25.0 (22.0)	16.0 (31.0)
CK (IU/L)	93.0 (80.0)	113.0 (569.5)
Neutrophil count ($10^{9}$/L)	3.2 (1.7)	4.7 (4.0)
hsTnI (ng/L)	2.4 (4.0)	77.2 (228.4)
Urea (mmol/L)	3.7 (1.5)	7.6 (3.3)
CRP (mg/L)	5.0 (17.0)	53.0 (56.5)
Hb (g/dL)	13.6 (2.0)	12.1 (3.3)
WBC ($10^{9}$/L)	5.2 (2.4)	7.1 (4.7)
Lymphocyte ($10^{9}$/L)	1.2 (0.8)	1.2 (0.7)
LDH (IU/L)	201.0 (64.5)	303.0 (95.0)
Creatinine (umol/L)	70.7 (27.0)	117.0 (42.0)
Platelets count ($10^{9}$/L)	216.0 (88.8)	204.0 (89.2)

**Table 2 diagnostics-12-02728-t002:** The distribution of various features according to Severity class based on the amount of oxygen consumed. The number of patients comorbidities in each class is indicated as such, while the median values of the test results and the interquartile range (in parentheses) are presented for each class.

	Stable	Mild	Serious	Critical
Age	44.0 (17.16)	64.5 (14.52)	60.0 (19.07)	66.0 (13.50)
Sex				
Men	155	20	6	22
Women	162	24	9	15
Underlying disease				
C.H.D.	5	1	0	2
Hypertension	25	9	2	6
Asthma	1	0	0	1
C.R.I	0	0	0	1
Diabetes Mellitus	23	5	4	2
H.M.	1	0	0	0
Test results				
Total bilirubin (umol/L)	7.0 (5.0)	7.0 (5.0)	7.5 (4.5)	9.0 (5.0)
ALT (IU/L)	24.0 (19.0)	24.5 (22.8)	33.0 (16.5)	38.0 (31.5)
CK (IU/L)	93.0 (74.0)	110.5 (84.0)	134.0 (243.0)	127.5 (228.0)
Neutrophil count ($10^{9}$/L)	3.1 (1.5)	3.9 (1.9)	4.2 (3.0)	4.6 (4.8)
hsTnI (ng/L)	1.9 (2.7)	6.3 (6.4)	15.0 (60.0)	16.7 (66.7)
Urea (mmol/L)	3.6 (1.6)	4.3 (2.2)	4.4 (3.7)	5.0 (4.0)
CRP (mg/L)	3.0 (10.0)	21.0 (51.5)	52.0 (84.0)	51.5 (78.8)
Hb (g/dL)	13.7 (2.0)	13.3 (2.2)	13.4 (2.6)	12.9 (2.3)
WBC ($10^{9}$/L)	5.1 (2.2)	5.6 (2.2)	5.7 (2.8)	6.3 (4.9)
Lymphocyte ($10^{9}$/L)	1.3 (0.8)	0.9 (0.5)	1.0 (0.3)	1.0 (0.6)
LDH (IU/L)	195.0 (54.0)	233.5 (140.8)	321.0 (173.5)	288.0 (148.0)
Creatinine (umol/L)	69.0 (26.0)	76.5 (29.5)	81.0 (29.8)	88.5 (49.8)
Platelets count ($10^{9}$/L)	218.0 (85.0)	199.0 (82.8)	207.0 (95.5)	201.5 (60.5)

**Table 3 diagnostics-12-02728-t003:** Contingency table for exposure diabetes mellitues to Severity of the case.

Diabetes Mellitus	Stable	Mild	Serious	Critical
No	302	37	10	35
Yes	19	7	5	3

**Table 4 diagnostics-12-02728-t004:** Contingency table for exposure hypertension to Severity of the case.

Hypertension	Stable	Mild	Serious	Critical
No	302	37	10	35
Yes	19	7	5	3

**Table 5 diagnostics-12-02728-t005:** Contingency table for exposure age to Severity of the case.

Age	Stable	Mild	Serious	Critical
≤25	42	0	0	0
(25,35]	64	0	0	0
(35,45]	63	6	1	0
(45,55]	51	5	0	6
(55,65]	59	11	6	12
>65	42	22	8	20

**Table 6 diagnostics-12-02728-t006:** Hypothesis testing results for Severity-outcome.

Feature	$\chi^{2}$	*p*-Value
Diabetes mellitus	18.41	0.0052
Hypertension	17.63	0.00036
Age	100.12	$1.23\cdot 10^{-14}$

**Table 7 diagnostics-12-02728-t007:** Contingency table for exposure chronic heart disease to Mortality.

C.H.D.	Alive	Dead
No	330	11
Yes	6	2

**Table 8 diagnostics-12-02728-t008:** Contingency table for exposure age to Mortality.

Age	Alive	Dead
≤25	36	0
(25,35]	49	0
(35,45]	69	6
(45,55]	54	5
(55,65]	73	2
>65	55	11

**Table 9 diagnostics-12-02728-t009:** Hypothesis testing results for the outcome Mortality.

Feature	Alive	Dead
C.H.D.	5.15	0.023
Age	39.1	$2.26\cdot 10^{-7}$

**Table 10 diagnostics-12-02728-t010:** The 8 features most associated with the two outcomes, as predicted by the mutual information classifier.

Mortality	Severity
Age	Age
Urea	CRP
hsTnI	LDH
Creatinine	hsTnI
LDH	Lymphocyte
Neutrophil	Creatinine
CRP	Urea
WBC	Neutrophil

**Table 11 diagnostics-12-02728-t011:** Accuracy of the ML algorithms to predict the outcome Severity.

	6 Features	10 Features	17 Features
GB	73.7%	74.5%	75.4%
DT	68.2%	68.8%	69.2%
RF	77.2%	76.6%	79%
RUSB	66.6%	73%	67.4%

**Table 12 diagnostics-12-02728-t012:** Accuracy of the ML algorithms to predict the outcome Mortality.

	6 Features	10 Features	17 Features
GB	96.2%	95.2%	96.2%
DT	95.5%	96.3%	96.2%
RF	96.2%	96.2%	96%
RUSB	95.2%	96.2%	95.2%

**Table 13 diagnostics-12-02728-t013:** Accuracy of the ML algorithms to predict the outcome Severity. SMOTE was used.

	6 Features	10 Features	17 Features
GB	78%	89.8%	92.7%
DT	74.2%	82.8%	83.6%
RF	80.1%	89.6%	94.6%
RUSB	51.3%	54.1%	56%

**Table 14 diagnostics-12-02728-t014:** Accuracy of the ML algorithms to predict the outcome Mortality. SMOTE was used.

	6 Features	10 Features	17 Features
GB	98%	98%	98%
DT	99.4%	98.4%	97.4%
RF	99.3%	98.5%	99.1%
RUSB	98.5%	98.5%	98%

## Data Availability

The datasets generated and/or analysed during the current study are not publicly available due to privacy agreement between the Queen Elisabeth hospital and its patient but are available, in symbolically encoded form, from the corresponding author on reasonable request.

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
