# Peer review of "Prognostic Model of COVID-19 Severity and Survival among Hospitalized Patients Using Machine Learning Techniques"

_diagnostics, 2022, doi:10.3390/diagnostics12112728_

Round 1

Reviewer 1 Report (Previous Reviewer 1)

1. Overall grammar and typographical errors can be checked

Author Response

Dear referee,

We are happy to note you did not have any major suggestions or changes.
Thank you for your work,

Ivano Lodato

Reviewer 2 Report (New Reviewer)

Major comments:

1.     It is unclear to me from the introduction what are the lacking’s exists in the recent cited study. And what kind of extra scientific significance exists in their articles compared to existing studies?

2.     From the methodology section, I am not clear how the authors select 429 patients for their observational study. How they are confident that size is enough to conduct their study? Have they used any kind of power test for conducting the study based on their sample size?

3.     The literature included in the introduction section not enough to understand the background of their research. I think they should include more recent studies in the introduction section with a logical sequence.

  1. It is necessary to rearrange the Discussion and conclusion part briefly with a more logical sequence according to their objective.
  2. It is not clear to me how this research may be helpful for the scientific community for future research.

Minor Comments:

1. Need to update the problem of solving grammatical issues such as articles, Voice, prepositions, and some sentence patterns.

Author Response

Dear reviewer,

please find attached a pdf file with our point-by-point response to your report.

Best regards.

Reviewer 3 Report (New Reviewer)

topic is good. If possible add any low cost COVID detection methods  in remote areas

Author Response

Dear referee,

We are glad to note you had no major suggestions or changes.
Thank you for your work,

Ivano Lodato

Round 2

Reviewer 2 Report (New Reviewer)

You have tried a lot to clarify my queries according to the first review. Thank you all authors for clarifying all of the doubts mentioned in the first review.

This manuscript is a resubmission of an earlier submission. The following is a list of the peer review reports and author responses from that submission.

Round 1

Reviewer 1 Report

1. Abstract needs to provide the machine learning algorithm that contributes better results.

2. Comparison of results with other Machine learning algorithms like Decision tree, k-NN, Naive Bayes, Aritifical Neural Network, Support Vector Machines etc., is not done. Why?

3. Justification to choose Random Forest classifier is required.

4. A tabular description of the datasets used, machine learning approaches and features used for COVID-19 prognosis need to be included in section 3.4

Author Response

Dear Referee,

thank you very much for your helpful comments and suggestions on how to improve our manuscript. Below, you will find our point-by-point answer to your report.
1. We have added more information about the ML classifiers used and the most accurate ones in the Abstract; we think the the addition should be sufficient.

2. Before answering to your question, we feel a short explanation is warranted: while it is true that a proper comparison of results between the ML algorithms we selected and the many other existing is missing, we did not think it was necessary or fitting, especially for the medical practitioners who may find our results useful, to give extensive (and more mathematical) explanations on all possible algorithms we could have used. Furthermore, for our analysis we have uses both ``old-fashioned'' statistical techniques and with 4 different ML algorithms (5 if one consider the mutual information algorithm (sec. 3.3) used for feature selection). For each run, i.e. each new training dataset, we select the algorithm(s) outperforming the others. This approach already seems quite broader than the average publication on the subject. 
To answer your question in more detail: Decision trees are used; the k-NN algorithm, although not explicitly mentioned or explained, is an integral part of the mutual information algorithm (see sec. 3.3) and it is used to deal with classification of mixed continuous and discrete/categorical variables. Naive Bayes and more generally neural network have not been used, simply because they require much larger datasets to outperform even a basic ML classifier such as decision tree; SVM are typically used for linear fitting and the non-linear (kernel based) approach again requires much larger datasets to perform efficiently. As it should be clear, we have considered, presented and used only algorithm allowing non-linear fitting, and for which the size of the dataset did not need be too large. Hope this answer is satisfying, but if more is required please do not hesitate to point it out.

3. Following the previous point, we did not choose Random Forest classifiers. In fact, in the graphs presented (Fig. 4 and Fig. 5) the random forest classifier was simply the one, among 4, with performed better to classify the Mortality and Severity of patients. If the use of Smote is considered, then also Gradient Boosting or RUS boosting perform very well.

4. This point you raise is not completely clear to us. Given the mention to section 3.4 ("Comparison with results from the literature") we initially believed you were asking for a table in which we present the results obtained in other prognosis studies for feature selection and compare it to our results. However, since this is not a review paper, we believe you actually wanted to suggest us to add a table where the 4 algorithms we used in this paper are compared. We added 4 tables (11-12-13-14) and a chunk of explaining test in section 3.5, outlining the average accuracies of all four algorithms over our dataset. Please do let us know if our interpretation of your suggestion is wrong, or only partially correct, we would be happy to add talbes, or figures, or re-organize the material already present, if it increases the readability and quality of our paper!

Thank you in advance for your further comments, 
we look forward hearing from you soon,

Regards,

Ivano Lodato and the authors.

Reviewer 2 Report

The work presented for review, "Prognostic model of COVID-19 severity and survival among hospitalized patients using machine learning techniques", undoubtedly concerns a very important topic. Unfortunately, I am sorry to say that in my opinion, it is not innovative enough to be published in Diagnostics. My opinion argues, inter alia, that the conclusions obtained in this work are only a repetition of those obtained in other works of this type published even in 2020. Patient studies took place in 2020, so we know that they do not concern diseases caused by the viral genotypes that are currently dominant. Moreover, the research group is small, especially in the case of this type of analysis and compared to the research of other authors.

Author Response

Dear Referee,

thank you for your report. We are sorry to say we disagree with your evaluation. You raise two main points: the innovative aspects of our paper, and the size of the dataset used, in your opinion too small. Concerning the latter point, most accepted publications we cite have analysed dataset of comparable, if not smaller size compared to ours (while at the same time, the number of features they used as input was bigger than ours).
Concerning your first point, it is indeed true that the data analysed in this publication have been collected 2 years ago, and we know that the currently dominant viral genotypes have changed since then. However, that is not what we would call innovation. In our paper, you will find standard statistical techniques as well as novel ones (at least to the best of our knowledge), meant to select features based on their distributions, that we created for this analysis!Furthermore, in our analysis we have also generated synthetic data to balance the originally imbalanced dataset, and presented a comparison between the two (to the best of our knowledge we are among the few groups who have done so).
Regardless of the techniques, simply the different ethnicity of the COVID-19 infected subjects, correlated to the distribution of their co-morbidities, demographics and blood-test results may alone produce innovative results, or at least results which have not been encountered in the literature. And if, as it is the case for many of our selected features, an overlap is found with other publications, we do not feel this result to be useless, or a simple repetition. Rather it is the signal of a common characteristics or indicative features related to COVID-19 which may be shared not only among the COVID-19 virus and all its variants, but also with future virus and seasonal flus. For these reasons, we feel it is important that the results of our paper, as well as other similar results, be published, so as to generate knowledge, regardless of whether the results are purely innovative or confirmational of certain statistical trends, whether they are recent or 2 years old. 
We would also like to point out that these data were released by the appropriate hospital authority after a long period of bureaucratic approval, and so by no fault of our own we have submitted this paper only now.

Certain of your familiarity with such problematics, and your understanding of the importance of publishing medical research papers, not only if their *final results* are innovative, we hope that you will re-consider your initial evaluation and look forward hearing back from you.

Best regards,
Ivano Lodato and the authors